# Genome Editing Strategies to Protect Livestock from Viral Infections

**DOI:** 10.3390/v13101996

**Published:** 2021-10-04

**Authors:** Jenny-Helena Söllner, Thomas C. Mettenleiter, Björn Petersen

**Affiliations:** 1Institute of Farm Animal Genetics, Friedrich-Loeffler-Institut, 31535 Neustadt am Rübenberge, Germany; jenny.soellner@fli.de; 2Friedrich-Loeffler-Institut, 17493 Greifswald, Germany; thomasc.mettenleiter@fli.de

**Keywords:** genome editing, gene editing, CRISPR/Cas, disease resistance, livestock viruses, host–pathogen interactions, viral interference

## Abstract

The livestock industry is constantly threatened by viral disease outbreaks, including infections with zoonotic potential. While preventive vaccination is frequently applied, disease control and eradication also depend on strict biosecurity measures. Clustered regularly interspaced palindromic repeats (CRISPR) and associated proteins (Cas) have been repurposed as genome editors to induce targeted double-strand breaks at almost any location in the genome. Thus, CRISPR/Cas genome editors can also be utilized to generate disease-resistant or resilient livestock, develop vaccines, and further understand virus–host interactions. Genes of interest in animals and viruses can be targeted to understand their functions during infection. Furthermore, transgenic animals expressing CRISPR/Cas can be generated to target the viral genome upon infection. Genetically modified livestock can thereby reduce disease outbreaks and decrease zoonotic threats.

## 1. Introduction

The current Covid-19 pandemic has highlighted the importance of potential zoonotic pathogens, which have become a primary focus for the life science community and the public. Therefore, the necessity of a ‘One Health’ strategy has become more apparent. Today, livestock farming in developed and developing countries facilitates the evolution and adaptation of pathogens to humans [1]. However, due to a steadily growing world population, livestock production must increase to cover the growing demand for animal products [2]. At the same time, animal production must become more sustainable to meet challenges such as climate change, antibiotic resistance, zoonotic outbreaks, and animal welfare [3]. One approach to increase sustainability is to control infectious disease outbreaks among livestock. Conventional efforts to prevent the spread of infectious diseases are based on biosecurity, involving surveillance, quarantine, immunization, culling, disinfection and hygiene, and the education of stakeholders [4]. Successful biosecurity implementations resulted in the local, national, regional or global eradication of pathogens. For example, Classical Swine Fever (CSF) and Foot-and-Mouth Disease (FMD) have been eliminated in most European countries, and peste des petits ruminants (PPR) might be globally eradicated by 2030 [5,6,7]. Rinderpest has already been eradicated globally since 2011.

Nonetheless, barriers may hinder the implementation of efficient biosecurity measures threatening livestock production [4,8]. Infectious disease outbreaks are often accompanied by the culling of large numbers of production animals, which threatens food security and livelihoods. Pig producers in Europe and Asia are currently challenged by African Swine Fever (ASF), while poultry holders fight against highly pathogenic Avian Influenza (HPAI). It is estimated that China lost about 30% of its pig population in 2019 due to ASFV, with numbers ranging from 150 to 200 million animals [9], though some unofficial estimates reach up to 50% [10]. The losses resulted in an 8% drop in global pork production in 2020 [11].

Between the 15th of January and the 4th of February in 2021, more than nine million poultry in Africa, Asia, and Europe were culled due to HPAI [12]. Protecting livestock from infectious diseases enhances animal welfare and reduces zoonotic risks, avoids the culling of millions of animals, and precludes significant economic losses in the agricultural sector. 

However, vaccines to combat viral diseases are not always available, and other biosecurity measures may fail. Therefore, traditional breeding strategies have also been adopted to develop breeding traits for disease tolerance or resistance. For instance, the immune response to Porcine Reproductive and Respiratory Syndrome Virus (PRRSV) varies strongly between individual animals and was considered for potential genetic selection [13,14,15]. However, many infectious diseases cannot be controlled by an improved immune response alone. Therefore, other strategies to supplement biosecurity and traditional breeding must be developed to protect animals and their producers from infectious disease outbreaks. A promising approach could be the recently developed direct genetic modification of livestock, which has evolved since 1985 [16]. Especially in recent years, targeted genetic modifications have become feasible due to the discovery and further improvement of genome-editing tools such as CRISPR/Cas9 [17]. These novel tools have already contributed to the generation of disease-resistant animals, paving a new way to prevent infectious disease outbreaks in livestock. 

PRRS-resistant pigs were generated by employing CRISPR/Cas9 technology to knock out the porcine CD163 receptor [18,19,20,21,22]. Transgenic expression of Cas9 rendered chickens resistant to Marek’s disease [23]. In addition, CRISPR/Cas9 systems have been employed to develop recombinant vaccines against Pseudorabies virus (PRV) or avian influenza [24,25]. Viral disease surveillance to prevent and control outbreaks can be assisted by CRISPR/Cas diagnostic tools, which have been created for various viruses [26,27,28,29]. This review will elaborate and discuss the possible contribution of the CRISPR/Cas systems to the development of disease-resistant animals and their potential for vaccine and diagnostic development. A literature search of relevant articles was carried out on scientific search engines, e.g., PubMed, using search terms such as: genome editing, genetic modification, gene engineering, disease resistance, livestock, poultry, cattle, pigs, CRISPR/Cas9, CRISPR/Cas diagnostics, CRISPR/Cas9 recombinant vaccines, and CRISPR/Cas screens. 

## 2. The Diversity of CRISPR/Cas

The most prominent additions to the genome-editing toolbox are CRISPR/Cas systems, which have been improved and adapted constantly since the discovery of the directed cleavage potential of Cas9 in 2012 [17]. CRISPRs (clustered regularly interspaced palindromic repeats) were first described in *E. coli* [30], but were later found to be a general part of the adaptive immune system of prokaryotes in combination with CRISPR-associated genes (*cas*) [31,32]. During the last decade, CRISPR/Cas systems have been studied and classified extensively [33,34,35,36,37]. They are divided into two classes where class 1 is subdivided into types I, II, and IV and into several subtypes, while class 2 consists of type II, V, and VI and their subtypes. Class 1 systems are characterised by a cascade of Cas proteins (cascade = Cas complex for antiviral defence) and a Cas endonuclease binding to a crRNA (CRISPR RNA) that is homologous to the target site. In contrast, class 2 systems are confined to one multidomain endonuclease protein, which binds to the crRNA. So far, mainly class 2 Cas proteins have been exploited for genome engineering purposes, although recently Cas3 of class 1 type I was also studied [38].

### 2.1. Cas9

Since the discovery of CRISPR/Cas9 as a genome-editing tool, it has evolved significantly. Cas9 is an endonuclease and is part of the class 2 type II CRISPR system that induces programmed double-stranded breaks (DSBs) in dsDNA. The enzyme contains two nuclease domains, the HNH and RuvC nucleases, which cleave the complementary and non-complementary strand, respectively. Cas9, as a genome-editing tool, was first recovered from Streptococcus pyogenes (SpCas9). So-called guide RNAs (gRNAs), consisting of a crRNA and a tracrRNA (trans-activating CRISPR RNA), were synthesised to induce DSBs at the desired location [17]. The discovery of CRISPR/Cas9 as a genome-editing tool is summarised in Doudna and Charpentier (2014) [39]. Since its first appearance, CRISPR/Cas has revolutionised genome editing due to its specificity and easy adaptation of gRNAs to fit research requirements, as compared to ZFN and TALEN. However, SpCas9 requires a 20 bp gRNA and a 5′ NGG 3′ PAM (protospacer adjacent motif) downstream of the gRNA for target recognition, thereby limiting the number of target sites. Nowadays, a variety of Cas9 versions are available with varying PAM sequences [40,41]. These alterations deliver a range of options to target almost any location in the genome of interest.

### 2.2. Cas12a 

While Cas9 was implemented as a systematic strategy to generate genomic edits, other Cas proteins were further explored. In 2015, Cas12a, formerly Cpf1, showed its potential as another genome-editing tool of the Cas family [42]. Cas12a belongs to class 2 of the CRISPR/Cas system, which targets T-rich PAM (5′ TTTN 3′) sequences, but does not require a tracrRNA. The crRNA recognises complementary strands, and Cas12a induces a DSB with its RuvC endonuclease domain. Diversification of target sites can be addressed by utilising the versatile Cas12a PAM sequence. The nuclease gives homology-directed repair experiments an advantage by leaving sticky ends after cleavage, allowing directed insertion [42].

### 2.3. Cas13 

In the meantime, another class 2 Cas protein was discovered, Cas13 (C2c2) [43]. Soon after its discovery, Cas13 of *Leptotrichia shahii* was found to cleave RNA [44]. However, after screening several orthologs, more effective Cas13 proteins were discovered [45,46]. Cas13 possesses two Higher Eukaryotes and Prokaryotes Nucleotide-binding (HEPN) domains that can be directed towards specific target RNA in combination with crRNA.

### 2.4. Cas3

The above-described CRISPR/Cas systems are all part of the class 2 CRISPR/Cas systems. The latest addition to the Cas toolbox is the CRISPR/Cas3 system, the first class 1 system redirected for genome editing. Class 1 systems are most common in archaea and less common in bacteria [33], and are composed of several Cas proteins, a cascade, and one endonuclease protein. Depending on the strain and subtype, the cascade of Cas3 varies. Deletions of up to 100 kb were reported by employing Cas3 [38,47].

## 3. Strategies to Protect Livestock from Viral Infection

Conventionally, the prevention of viral infections is managed by biosecurity measures to limit the spread of pathogens [4,48]. These measures include disinfection and hygiene as well as surveillance, quarantine, vaccination, culling, and the education of stakeholders [4]. Due to these measures, the threat of infectious diseases can be reduced. Although, many infectious diseases are presently controlled through vaccination programs, there are limitations. For instance, ideal animal vaccines should be inexpensive and easy to apply to the target species, and preferably, should be effective after a single dose, capable of mass application, and induce a solid immune response that mimics natural infection, as summarized for avian influenza virus (AIV) vaccines by Swayne and Sims (2021) [49]. Differentiation between infected and vaccinated animals (DIVA) by genetic or serological markers is crucial for surveillance of the spread of diseases. If vaccination is recommended, other constraints such as production, logistics, and biosecurity must be considered [49]. When mass vaccination is applied, elimination of the disease mainly depends on the vaccine coverage and efficacy. For example, vaccines are widely used to control PRRS, but due to their low effectiveness, eradication of PRRSV is not realistic [50]. Still, no vaccines are as-yet available for other infectious diseases, such as ASF. There is, therefore, a need to develop alternative or complementing strategies for infectious disease prevention and control. 

Genome editing strategies provide great potential to complement classical disease control or eradication measures. The most obvious contribution is to generate livestock that is resistant to diseases by genetically modifying the host genome. Other strategies may integrate Cas9 as a transgene into the host’s genome and target essential genes of the invading virus from within the animal. In addition, the CRISPR/Cas system can be employed to study host–virus interaction in vitro.

### 3.1. Mechanisms of Genome-Editing Tools

Genome-editing tools such as CRISPR/Cas, TALENs (transcription activator-like nucleases), and Zinc-finger nucleases induce DNA double-strand-breaks at specific locations within an organisms’ genome. DNA is cleaved by a specific endonuclease and then repaired by either of two mechanisms: non-homologous end-joining (NHEJ) or homology-directed repair (HDR) (Figure 1) [51]. NHEJ, the most common—but also a highly error-prone—repair mechanism, generates insertions and deletions (indels), thereby frequently causing frameshifts and producing functional knockouts (KO). On the other hand, HDR is a rare mechanism since it only occurs after DNA replication in the late S and G2 phase [52], but it can facilitate knock-in (KI) strategies. HDR modifies the targeted site by providing a DNA donor template flanked by arms that are homologous to the adjacent sequence (Figure 1). Depending on the objective, either to insert a gene or to inhibit gene function, HDR or NHEJ is preferred, respectively. TALENs and Zinc-finger nucleases are known for their limitations, such as available target sites [53], target specificity [54], and laboriousness [55]; hence, CRISPR/Cas was rapidly adapted for most genomic engineering experiments.

### 3.2. CRISPR/Cas as Viral Interference

Before genome-editing tools became available to address viral resistance in animals, RNA interference (RNAi) was employed to generate resilient animals. RNAi refers to post-transcriptional gene silencing by short interfering dsRNA (siRNA) and was first discovered in 1998 in C. elegans [56]. RNAi has provided a tool that is engineered to target RNA viruses in animal cells [57,58,59,60], which is limited to post-transcriptional viral interference and, therefore, only acts when viral replication has already proceeded. Another new strategy, called in vivo pathogen genome targeting [61], has been developed since the discovery of the CRISPR/Cas9 system as a genome-editing tool (Figure 2). The strategy of integrating Cas9 and virus-specific gRNAs into the host genome, and thereby targeting the viral genome ahead of replication, has been successful in vitro in human cells [62], in Nicotinia benthamiana plants [63,64], and in the silkworm Bombyx mori [61].

### 3.3. Elimination of Genetic Susceptibility

The most straightforward approach to generate resistant livestock is to limit their susceptibility to viruses [18]. Genome-editing tools such as CRISPR/Cas made it possible to target specific genes of animals that are vulnerable to viral infection. Receptors responsible for viral entry can be eliminated to protect the animals from infection. In addition, genetic inter/intraspecies variation, which is accountable for resistance, can be exploited and targeted KI or KO can facilitate resilience against viruses (Figure 3).

#### Gene Drives

Gene drives can potentially also contribute to the generation of resistant livestock or assist in controlling viral vector-borne infections. The inheritance of modified alleles can be promoted in combinations with CRISPR/Cas gene drive alleles. As demonstrated in several potential vector insects such as mosquitos [65,66], vector species transmitting animal diseases could be modified to decrease their population. The mechanism of synthetic gene drives is based on natural occurring site-specific homing endonuclease genes (HEGs) [67]. HEGs recognize site-specific sequences in the homologous chromosomes that are not carrying the HEG and induce DSBs. The break in the homologous chromosome is repaired via homologous recombination using the HEG-containing chromosome as a repair template that integrates the HEG sequence, a process called ‘homing’. HEGs thereby turn heterozygous alleles into homozygous alleles, and inheritance is promoted [67]. Similarly, the endonuclease Cas and its gRNA can be integrated into a target site to create a ‘CRISPR allele’ [66], which is programmed to induce DSBs in desired genomic locations and, in the meantime, copy the system onto the wild-type chromosome.

## 4. Disease Resistance in Livestock

Long before genome editors were known, the vision of disease-resistant livestock to improve animal production existed within the scientific community (Table 1). However, only after RNAi and genome editors were discovered, were the first breakthroughs reported.

### 4.1. Random Integration of Transgenes

Pigs were among the first animals that were genetically modified to resist viral infections. Weidle et al. (1991) generated rabbits and pigs that expressed mouse monoclonal antibodies encoded by Ig heavy and light chain genes, which would act as a potential general in vivo immunization against, e.g., influenza [84]. In the same year, Lo et al. (1991) reported the integration of mouse IgA chains into sheep and pig genomes, pursuing a similar goal. No IgA translation was detected in most animals except for one founder sheep, which expressed mouse IgA in peripheral lymphocytes [85]. The following year, a possible resistance towards influenza virus in pigs was reported by transferring the murine myxovirus-resistant (Mx1) system into the porcine genome. Nonetheless, this study also suffered from inefficient synthesis of the foreign protein [68].

Another more specific attempt was made to confer resistance against Visna Virus, an ovine lentivirus that causes pneumonia, arthritis, and encephalitis in sheep [83]. Viral envelope genes were inserted into the sheep genome through microinjection, generating three living lambs that expressed viral envelope mRNA in vitro. Two of them subsequently produced antibodies against the foreign protein. mRNA and envelope proteins were also found in several tissues such as the kidney, muscle, and brain [86].

### 4.2. RNA Interference

RNAi was used in chickens against Marek’s disease virus (MDV) [81] and avian influenza viruses [57], targeting genes that are essential for viral replication. By targeting the MDV glycoprotein B and infected-cell-polypeptide-4 (ICP-4) genes with RNAi, viremia and the occurrence of clinical signs were significantly reduced [81]. RNAi targeting the expression of AIVs polymerase showed no differences in directly infected transgenic and wild-type chickens in terms of disease development, but virus transmission was reduced. Infected transgenic (TG) and control chickens were each housed with uninfected wild-type (WT) chickens. The TG group only transmitted to 2 out 10 co-housed birds compared to the control group, in which 7 sentinel animals died [57]. Foot-and-mouth disease is a devastating disease in cloven-hoofed livestock caused by FMDV. Wang et al. (2012) generated bovine foetuses expressing a shRNA (short-hairpin RNA) that targeted viral protein (VP) 4 of FMDV [58], a highly conserved protein in different FMDV strains [87]. Tongue epithelium cells of these foetuses were infected with FMDV and were found to have a 91% reduction in viral replication in vitro [58]. In 2015, resilience against FMD was first shown in modified pigs. Transgenic pigs expressing siRNAs were produced to interfere with the expression of viral protein 1. The animals showed a stable integration of the transgene, but the expression levels varied widely between individual animals and within the animals’ tissues. Therefore, the pigs were divided into low and high siRNA-expressing groups before the viral challenge. Low-expressing animals experienced mild fever (39.5–40 °C) and some lesions on day 7 post infection (dpi). The high-expressing pigs showed no signs of increased temperature, but all developed one minor vesicle at 9 dpi.

In comparison, the control group showed clinical signs such as lesions 3 dpi. As expected, the viral load in serum and lesions was also lower in the TG groups than in the control group. Thus, although the pigs were not entirely resistant to clinical signs of the disease, lesions and viral load were significantly reduced [69].

While most of the aforementioned viruses are of concern for animal production, porcine endogenous retrovirus (PERV) is considered a potential threat for the application of pigs as a source of xenotransplants. PERVs are integrated in the porcine genome and can translocate into the human genome in vitro [88], although in vivo integration has not yet been observed. In pigs, PERVs do not cause any clinical diseases but may be harmful in humans. Observations of endogenous retrovirus of different species and the preferred integration sites of PERVs in the human genome suggest that PERV may induce tumorigenesis [88,89]. Transgenic pigs expressing shRNAs that targeted PERVs were generated [78,79], and the expression of PERV was reported to be reduced by up to 94% [79].

### 4.3. In Vivo Pathogen Genome Targeting

An in vivo pathogen genome-targeting strategy was directed towards MDV in chickens. Cas9 and gRNAs targeting the gene encoding the viral ICP-4 protein were integrated into the genome by a Tol2 transposon. The combination of transgenes successfully inhibited viral replication by inducing 1–20 bp deletions in ICP-4 [23]. In vivo pathogen genome-targeting was also employed to target CP204L, the gene encoding p30 of ASFV. P30, a phosphoprotein, is involved in viral entry and internalization [90]. Targeting the p30 encoding CP204L gene of the ASFV strain Armenia 2007 in a Cas9- and gRNA-expressing wild boar lung cell line resulted in a significant reduction in viral replication upon infection [91]. Based on these results, we generated Cas9- and gRNA-expressing pigs and infected transgenic and control animals with ASFV Armenia 2007. Despite the promising results from the in vitro study, no differences in ASFV infection were observed (unpublished). Targeting of the PRRSV genome was successful in MARC-145 cells. Cas13b was programmed to knock down the expression of ORF5 and 7 of PRRSV. In vitro, PRRSV infections exhibited > 90% reductions in targeted RNA [92].

### 4.4. Elimination of Susceptibility

The generation of gene-edited pigs that were fully resistant against the viral disease PRRS, using CRISPR/Case9, was reported in 2016. In piglets, PRRSV causes respiratory symptoms while inducing abortions and stillbirth in gestating sows [93]. Economic losses by PRRS are estimated to be as high as 664 million dollars per year in the United States [94] and 1.5 billion in the European Union (EU) [95]. Three pigs partially lacking the porcine CD163 receptor after CRISPR/Cas9-induced gene editing were challenged with PRRSV and did not develop clinical signs, anti-viral antibodies, or viremia [18]. The CD163 receptor, which allows PRRSV entry, is expressed by mature macrophages [96]. CD163 acts as a fusion receptor, particularly its fifth scavenger receptor cysteine-rich (SRCR 5) domain [97]. However, CD163 has also been shown to have important biological functions such as anti-inflammatory regulations by removing hemoglobin from blood plasma preventing oxidative toxicity [98,99]. After reports of PRRSV resistant CD163 KO pigs appeared [18], efforts were made to reduce the KO to the SRCR 5 domain [20,21,90] or to replace it with a CD163-like homolog [19] to maintain physiological function. These improved strategies to minimize KO trade-offs proved to be effective [19,20,21,22,71].

In the meantime, the crucial receptor for transmissible gastroenteritis virus (TGEV), an alphacoronavirus that causes diarrhoea and dehydration, has also been identified through a targeted KO. Pigs with an aminopeptidase N (ANPEP) KO were generated [72,73] and infected with TGEV and porcine epidemic diarrhoea virus (PEDV). The pigs showed no signs of infections with TGEV but remained susceptible to PEDV. These studies confirmed ANPEP as the physiological receptor for TGEV. Xu et al. (2020) generated pigs that carried a double KO for CD163 and ANPEP. The edit rendered the pigs resistant to PRRSV and TGEV while maintaining production and reproduction performances [71]. In addition, the susceptibility to porcine delta coronavirus (PDCoV) was investigated. PDCoV was discovered in 2012 [100], and its entry and replication mechanism has not been fully revealed [101]. The involvement of ANPEP as an entry receptor for PDCoV was indicated [102]. However, when CD163 and ANPEP KO pigs were challenged with PDCoV, resistance to the infection was only limited. Antibodies (AB) were not detected at 7 dpi but were detected at 14 dpi in KO animals. At 14 dpi, AB levels showed no differences compared to the WT animals.

Nonetheless, KO animals did not experience mesenteric hyperemia or thinning of the small intestinal wall, although no differences in small intestinal lesions were observed between KO and WT animals. In addition, the infection of porcine alveolar macrophages of KO and WT pigs indicated a significant decrease in susceptibility. Therefore, the study stated that ANPEP plays a role in viral replication but is not the sole mechanism governing PDCoV infection [71].

Other studies yielded similar results after generating pigs with genetic modifications that are thought to be responsible for disease development. Due to complex virus–host interactions during infection, single-gene edits may, in many cases, not be sufficient to support resistance against viral diseases. For instance, infection with ASFV appears to be too complex to be reduced to a single genetic mechanism, although the severity of the clinical symptoms of ASF is most likely linked to genetic variation. In domestic pigs (Sus scrofa domesticus) and wild boar (Sus scrofa ferus), the disease causes up to 100% mortality [90], whereas warthogs (Phacochoerus africanus) and African bush pigs (Potamochoerus porcus) only experience persistent subclinical infections [103,104]. One candidate gene that might be responsible for the disease severity of ASF is the RELA (v-rel reticuloendotheliosis viral oncogene homolog A) locus. The locus displays variations in three amino acids (aa) between the domestic pig and warthogs [105]. The viral gene A238L encodes a protein that is partially homologous to porcine IκBα [106], acts as a substitution for IκBα, and binds to RELA (p65), an NF-κB dimer [107]. The transcription factors of the NF-κB family are essential to maintain immune homeostasis and T- and B-cell functionality, and to regulate certain proinflammatory cytokine expressions [108]. ASFV infection inhibits the degradation of IκB, resulting in the absence of NF-κB in the cell nucleus blocking the NF-κB pathway [107]. By employing Zink-finger nucleases, three RELA-modified piglets were generated. Two carried modifications at all three aa (3aa) locations, and one pig carried modifications at only two aa (2aa) locations [109]. These animals served as founders to generate offspring that were homozygous for 3aa or 2aa alterations to be subjected to ASFV infections. Unfortunately, no resistance to ASF was observed. However, animals that carried a 3aa modification showed delayed clinical symptoms, and their viral load in blood and nasal secretion was decreased [76]. Although the entry mechanisms of ASFV is not yet fully understood, ASFV primarily infects macrophages. Since CD163 KO pigs showed resistance to PRRSV, and previous research identified CD163 as an ASFV receptor [110], CD163 KO pigs were also challenged with ASFV. However, no differences were observed after in vitro or in vivo infection, excluding CD163 as a relevant entry protein for ASFV [77].

CRISPR/Cas9 was also employed to disrupt all integrated copies of PERV [80]. Pigs were generated by somatic cell nuclear transfer (SCNT), in which all copies of PERVs were inactivated, preventing viral transmission from pigs to humans. The genome-edited pigs remained healthy, and karyotyping did not reveal any abnormalities.

Recently, CRISPR/Cas9 has been used to render chickens resistant to Avian Leukosis Virus subgroup J (ALV-J) [82]. ALV-J infects domestic chickens, turkeys and jungle fowls, and induces tumour formation [111]. Interestingly other galliform birds are not susceptible due to interspecies variation. The NHE1 (Na+/H+ exchanger type 1) receptor in chickens differs in terms of aa composition (W38), making the chicken susceptible to ALV-J [112]. By employing CRISPR/Cas9, the coding sequence of W38 was deleted in primordial germ cells (PGCs), and W38 +/− and −/− chickens were generated. In vivo infection studies revealed that homozygotic KO chickens, but not heterozygotic chickens, were resistant to ALV-J [82].

## 5. Discussion

### 5.1. Eliminating Susceptibility

Different strategies for protecting livestock from viral diseases with CRISPR/Cas have been described (Figure 4). The diversity of CRISPR systems for genome engineering continues to evolve, increasing the number of potential target sites, reducing off-target effects, and expanding effectiveness against DNA and RNA viruses. CRISPR/Cas can alter host genomes to prevent infection or control viral multiplication by means of in vivo pathogen genome targeting.

The first successful generations of resistant livestock have been reported in recent years. Potential adverse effects of knocking out target genes within the host must be evaluated, for instance, limiting the CD163 KO to the SRCR 5 domain in PRRSV-resistant pigs [20,21,90] to avoid immunological dysfunction [98,99]. In addition, before generating resistant animals, solid groundwork must be undertaken to identify target genes. CRISPR/Cas screens can identify relevant genes in the pathogen or the host. Genome-wide CRISPR screens are widely applied to study pathogen–host interactions [113,114,115]. The results of those screens can highlight genes in the host or virus genome that are critical for viral infection and/or multiplication in order to further understand viral infection or deliver strategies for targeted genome edits. CRISPR/Cas9 screens recently revealed several host genes that are essential for infection with Japanese encephalitis virus (JEV) by establishing a porcine genome-scale CRISPR/Cas9 knockout library [116]. JEV is a mosquito-borne virus and infects mammalian species, including humans and pigs. Similarly, sphingomyelin synthase 1 was identified as an entry factor for PRV [117], of which pigs are the natural host. Both screens used more than 80,000 gRNAs to target host genes in order to understand viral entry mechanisms.

After identifying targets, in vivo experiments may not result in resistant animals, as shown for ASF [77]. Inter- and intraspecies variation can also be the underlying cause of disease resistance or tolerance. Further investigations of genetic variations between animals may eventually identify the responsible genetic variation for the course of the disease. Genome-wide association studies (GWAS) are a tool to identify genetic variation between animals associated with genetic traits and are frequently used to generate breeding traits for genomic selection. GWAS could indicate inter/intraspecies variants that are accountable for variations in immune responses towards diseases such as PRRS or ASF. So far, 2932, 856, and 609 genomic regions, so-called quantitative trait loci (QTL) for disease susceptibility, have been identified in cattle, chickens and swine, respectively [118]. Genomic selection was implemented in modern livestock breeding programs based on QTLs and their underlying quantitative trait nucleotides (QTNs) [119]. Genomic selection will increase the number of validated QTNs for traits, but genetic improvement will be limited to biological constraints such as recombination. Therefore, Jenko et al. (2015) proposed the ‘Promotion of alleles by genome editing’ (PAGE) [120]. Their model calculated the response to selection by combining genomic selection and PAGE in breeding sires, which resulted in an enhanced response to selection compared to genomic selections alone. However, since many QTNs affect quantitative traits such as immune response, it will be challenging to introduce these modifications simultaneously. With further developments of genome-editing tools such as CRISPR/Cas, it might eventually be possible to combine conventional breeding and genome editing in livestock [120].

Furthermore, gene drives have the potential to be combined with conventional breeding. A gene drive induced by CRISPR/Cas was experimentally employed in mosquitos to reduce vector-borne diseases such as malaria [66]. Despite the beneficial effect of gene drives in terms of reducing the spread of disease, the eradication of these vectors could have unforeseen ecological consequences. Genetically modified vectors released into stable ecological systems will have to be closely monitored to avoid undesired developments. Promoting the inheritance of modified disease-resistant alleles in livestock, on the other hand, would be less difficult to control since livestock production is usually a closed system. By taking necessary precautions, the risk of gene drives escaping into the wild population would be limited.

When disease resistance can be linked to a single gene, resistant animals can be generated via genome editing and be incorporated into breeding programs [121]. For example, for PRRS-resistant pigs, the control of PRRSV is limited due to the moderate efficacy of inactivated vaccines and the safety risk of modified live virus vaccines [122]. The preliminary model of Petersen et al. (2021) suggests that a combination of genome editing and a mass vaccination program would eliminate PRRS in three to six years. In one scenario, the group calculated how many genome-edited pigs would be required to eliminate PRRS at a national level. The model showed that when all farms are exposed at an average reproductive rate of 1.5 of PRRSV and the vaccine is 70% effective, only 12% of the pigs would be need to be resistant to eliminate the disease. Hence, genome editing can potentially facilitate disease eradication [121].

### 5.2. In Vivo Pathogen Genome Targeting

The CRISPR/Cas system has great potential for in vivo pathogen genome targeting. Recently, we generated pigs carrying Cas9 and a gRNA-targeting ASFV and observed no abnormalities in the animals (unpublished). Cas9-expressing pigs were healthy and showed no reproductive impairments [123,124]. In addition, no toxicity was observed in transgenic chickens expressing Cas9 and gRNAs against MDV [23]. Although no direct studies on animal health and the performance of Cas9-expressing animals have been conducted, it can be deduced that Cas9 and gRNA expression, targeting ASFV and MDV, does not affect animal health.

As a genome-editing tool, Cas proteins have been shown to produce off-target effects. However, systems have been developed to address these concerns, e.g., modified Cas proteins to enhance specificity. Nonetheless, the genomic integration of Cas-encoding genes most likely results in a constant expression of Cas in the animals. Prolonged Cas expression may result in unwanted off-target activity even with engineered Cas proteins. CRISPRi (CRISPR interference), similar to RNAi working at the transcriptional level, may ease the concern of off-target cleavage. CRISPRi employs a dCas, a nuclease-dead Cas that is incapable of inducing DSBs but still binds to its target sequence, and thereby hinders transcription [125]. However, when concerned about off-target cleavage within the genome, off-target disruption of transcription may just be as harmful as inducing indels. For the CRISPR/Cas system to be considered a legitimate viral interference, Cas expression must be controlled. For example, one promising study by Oakes et al. (2019) was to install an ‘on-off switch’ within dCas9, which becomes activated by an endogenous signal [126].

A circular permutation rearranged the termini of Cas9 to control its activity and DNA binding. In addition, protease AA linkers between the N and C-terminal were conjugated to Cas9, resulting in the inactivation of Cas9. After exposing Cas9 and its protease linker to the sequence-specific protease, proteolytic cleavage of the linker sequence occurred, and Cas9 activity was restored. The engineered version was named ProCas9 and functioned with several viral proteases such as Zika virus (ZIKV) and West Nile virus (WNV). It was found that after Cas9 is ‘switched back on’, it would still have to find its target and disrupt viral replication. However, viral replication of Flaviviruses such as ZIKV and WNV was shown to be rapid, and genomic viral disruption may be ineffective. Hence, the authors proposed an alternative strategy of inducing cell death. Guide RNAs targeting, e.g., repetitive sequences or essential genes in the cellular genome would induce programmed cell death, thereby interrupting viral replication [126]. This strategy would also avoid potential viral escape mutants. Several studies have shown that in vivo targeting of human immunodeficiency virus 1 (HIV-1) can result in infectious escape mutants [127,128,129]. Indel formation within the Cas9 target sequence could give rise to competent viral strains, though time-bound differences were observed when targeting conserved or non-conserved domains [129]. It is, therefore, desirable to induce DSBs at several conserved locations of the viral genome or generate deletions in order to avoid potential escape mutants [127].

### 5.3. Ethical Justification of Genome Engineering Animals for Disease Resistance

Whether genome engineering in animals can be ethically justified depends on many reasons and the purpose of the genetic modification. A detailed systematic overview is given in de Graeff et al., (2019), incorporating arguments for and against the genetic modification of animals [130]. While this review focuses on engineered disease resistance, some arguments for and against it will be elaborated. First, disease resistance in animals will decrease the suffering of animals, thereby increasing welfare [131]. Secondly, the development of CRISPR/Cas provides safer methods to induce genetic modifications and limit off-targets [132]. Due to its precision, the success rates of creating the desired genotype have increased, and therefore, less animals with undesired genotypes would be produced [133,134]. Third, the animals’ general health could be improved by removing recessive alleles that cause health deficiencies [135]. It may be argued that enhanced disease resilience would lead to intensification of farming systems; however, the current trend in the livestock industry increasingly focuses on animal welfare rather than intensification [136].

Nonetheless, there are valid ethical reasons against targeted genome modifications. Despite the enhanced accuracy of CRISPR/Cas tools, there are still risks of off-target mutations, which could have a negative effect on animal health [132]. Generating modified offspring will depend on embryo technologies such as SCNT, which is associated with birth defects and postnatal deaths and, therefore, jeopardizes animal welfare [137]. In addition, depending on the modification, the intrinsic value or dignity of the animal may be compromised or lead to unwanted side effects when altering the immune system [18,135,136]. There is still room for ethical debates on whether animals should be modified in general and for disease resistance specifically, especially now that tools are available that are safer and easier to apply.

### 5.4. Regulation of Genome-Edited Livestock

While CRISPR/Cas systems pave novel ways to breed for improved animal welfare and contribute to sustainable livestock production, governmental regulations hinder research and development and, hence, gene-edited livestock, especially in the European Union (EU). The EU recently published their study on ‘new genomic techniques’, which was conducted due to the ruling of the European Court of Justice in 2018, in which it was stated that gene-edited products fall into the definition of genetically modified organisms (GMO). The report confirms the decision but highlights the fact that the European Food Safety Authority considers some cases of genome edits to have the same risk as conventional breeding, and thus, that the potential of genome editing for sustainable agricultural production needs to be further evaluated [138]. Countries in Latin America (Brazil, Argentina, Chile, Paraguay and Uruguay) decided to assess genome-edited products on a case-by-case basis [139]. In the United States, progress has been made in crop production by permitting CRISPR/Cas-modified button mushroom (*Agaricus bisporus*) to receive a non-regulated status in 2016 [140]. Since then, other crops, e.g., corn, followed. In addition, two transgenic animal species, salmon [141] and pigs [142], have been approved for consumption by the U.S. Food and Drug Administration, though these animals were not created with genome editors.

### 5.5. CRISPR in Disease Control and Prevention

While ethical justifications and approval regulations for genome-edited livestock are still being discussed, CRISPR/Cas assists in fundamental virology research and vaccine development (Figure 4). Editing of large DNA viruses as potential vector vaccines has been made possible. Adenoviruses, herpes simplex virus 1 [143], vaccina virus [144], PRV [145,146], and herpesvirus of turkeys [25,147], for example, have been edited to explore the Cas9-mediated homologous recombination of transgenes. Several Cas9-edited vector vaccines have been designed to induce immunity against poultry and livestock diseases. Poultry viruses such as avian herpes virus, duck enteritis virus, infectious laryngotracheitis virus, or Marek’s disease virus offer the possibility to generate multivalent vaccines for common viral infections [25,147,148,149,150,151]. For instance, Tang et al. (2020) developed a recombinant vaccine by inserting H9HA of AIV and glycoprotein D-glycoprotein I of infectious laryngotracheitis virus into a recombinant avian herpesvirus strain that expressed VP2 of infectious bursal disease virus [25,147]. Moreover, PRV is a double-stranded DNA herpes virus, and Cas9 can be readily applied to generate bivalent recombination vaccines. An attenuated PRV vaccine, which was missing glycoprotein E and the thymidine kinase, recombined with the gene that encoded the major capsid protein of porcine circovirus type 2 (PCV2), protected PRV in mice and pigs and induced sufficient levels of neutralizing antibodies for PCV2 [145]. Attenuated deletion vaccines can also be rapidly produced with CRISPR/Cas9 by inactivating virulence-determining genes [24]. An attenuated PRV vaccine was developed targeting the genes that encode nonessential glycoproteins E and I, and thymidine kinase. The vaccine was able to protect mice from developing clinical symptoms associated with PRV infection [24]. More details of CRISPR/Cas9 applications in vaccine development are provided by Tang et al. (2021) [152].

Monitoring viral diseases in livestock is essential to control and prevent outbreaks. Several diagnostic tools based on CRISPR/Cas have been developed. SHERLOCK [153], HOLMES [26] and DETECTR [154] are the most prominent examples of CRISPR-based viral detection methods that rely on the *trans*-cleavage potential of the Cas12a and Cas13a proteins. HOLMES was able to detect RNA viruses such as JEV and also DNA viruses, e.g., PRV. In addition, the system could differentiate between various strains of PRV and JEV [26]. SHERLOCK, based on Cas13a RNA cleavage and isothermal amplification, was adapted to detect PRRSV of clinical samples in lateral flow detection [155]. Several on-site diagnostic tools for ASFV have been developed utilizing Cas12a cleavage [27,28,29] and Cas9 [156]. According to the reports, the methods displayed high sensitivities and specificities that were comparable to state-of-the-art diagnostics, thus delivering new possibilities for disease surveillance.

## 6. Conclusions

In conclusion, CRISPR/Cas systems can be helpful in the development of vaccines and diagnostic tools to control and monitor viral outbreaks. By directly editing animal genomes, host–virus interactions can be further studied and animals that are resistant to viral infections can be generated. Thereby, zoonotic risks decrease, animal welfare improves, and economic losses are minimised. The research and development of CRISPR/Cas applications in livestock has developed fast, improving the safety of genome-edited animal products. Policies and regulations around the globe must adapt to scientific progress and address the potential of genome-edited livestock to contribute to a more sustainable future.

## Figures and Tables

**Figure 1 viruses-13-01996-f001:**
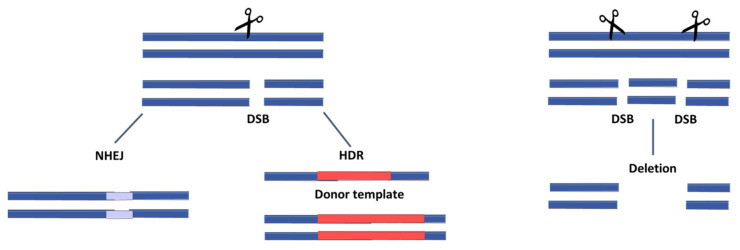
Genome editing strategies: Genome editors induce targeted double-strand breaks (DSB). Single target sites can be repaired by either non-homologous end joining (NHEJ)-generating inserts and deletions or by homology-directed repair (HDR). By providing a donor template, the desired sequence can be integrated via HDR. Large deletions can be induced by cleaving two target sites in the locus of interest.

**Figure 2 viruses-13-01996-f002:**
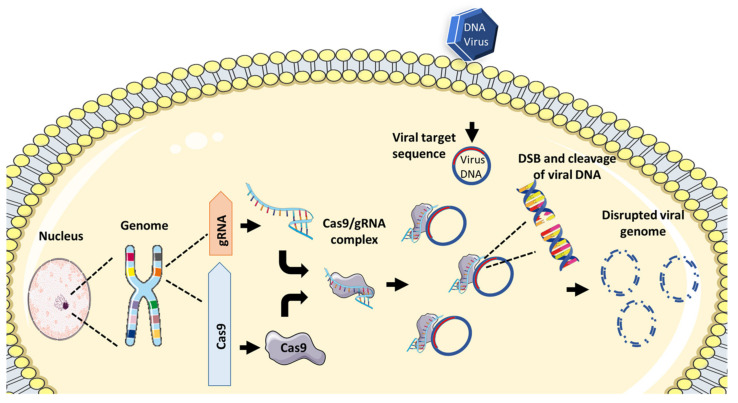
In vivo pathogen genome targeting: Cas9 and gRNAs (guide RNAs) targeting virus genomes are integrated into the host genome via genome editors or transposon systems. Cas9 and gRNAs are expressed and form a gRNA-Cas9 duplex. The gRNAs target the complementary DNA of the virus genome, and Cas9 induces a double-strand break (DSB) blocking viral replication.

**Figure 3 viruses-13-01996-f003:**
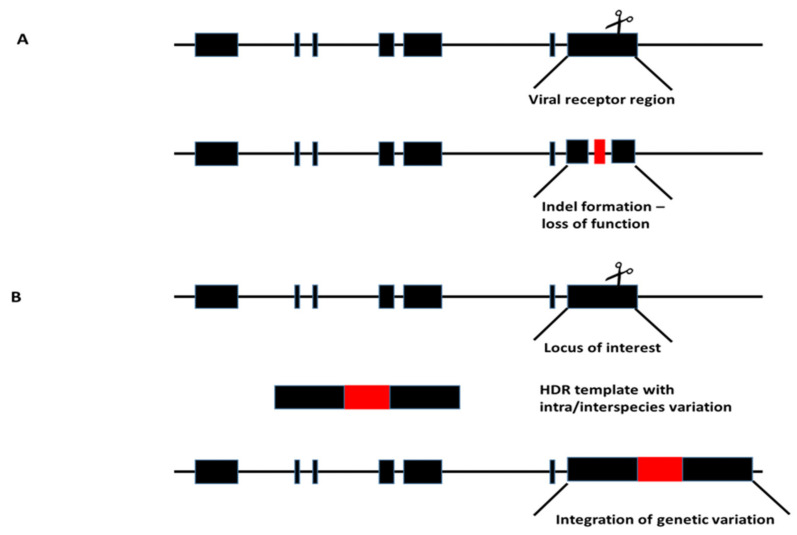
Targeting loci of interest: (**A**) Previously identified receptors susceptible to viral infection could be targeted with genome editors, thereby inducing loss of function. (**B**) Identified inter/intraspecies variations responsible for viral susceptibility can be inserted in the locus of interest by homology-directed repair (HDR).

**Figure 4 viruses-13-01996-f004:**
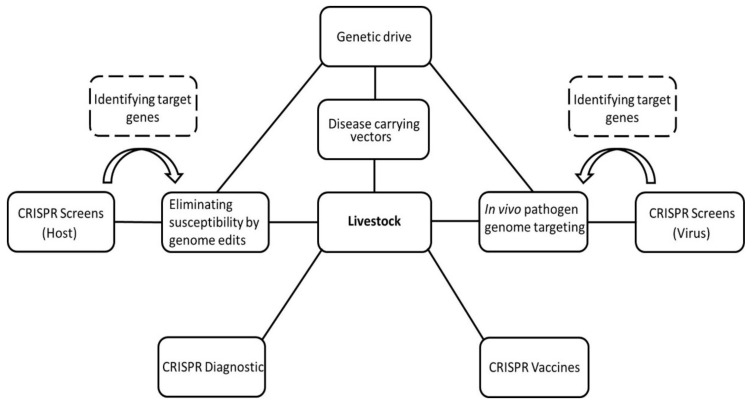
Genome editing as a tool to protect livestock from viral infection: Viral resistance in livestock can be achieved with CRISPR/Cas by either eliminating susceptibility or in vivo pathogen genome targeting. Target genes of the host organism or virus can be identified with CRISPR screens. Gene drives can promote the inheritance of the modified alleles or be integrated into vector species to disrupt transfection cycles. CRISPR/Cas-based diagnostic tools and vaccines can help to prevent and control viral outbreaks.

**Table 1 viruses-13-01996-t001:** Studies investigating viral resistance/resilience in livestock.

Virus	Gene		SCNT/ Micromanipulation	Method	Reference
	Host	Virus			
**Pigs**					
Influenza viruses	Mouse Mx1		PNI	DNA construct	[68]
FMDV		Nonstructural protein 2B, Polymerase 3D	SCNT		[60]
		Viral Protein 1	SCNT	RNA interference	[69]
PRRSV	CD163		SCNT	CRISPR/Cas9	[18]
	CD163 SRCR5		CMI	CRISPR/Cas9	[20]
	CD163-like homolog		SCNT	CRISPR/Cas9	[19]
	CD163		SCNT	CRISPR/Cas9	[21]
	CD163 SRCR5		SCNT	CRISPR/Cas9	[22]
CSFV		NS4B	SCNT	CRISPR/Cas9 and RNA interference	[70]
PRRSV, TGEV, PDCoV	CD163 SRCR5 and ANPEP		SCNT	CRISPR/Cas9	[71]
TGEV, PEDV	ANPEP		CMI	CRISPR/Cas9	[72]
	ANPEP		SCNT	CRISPR/Cas9	[73]
PEDV	CMAH		CMI	CRISPR/Cas9	[74]
ASFV	RELA		CMI	Zinc-finger nucleases	[75,76]
	CD163		SCNT	CRISPR/Cas9	[77]
PERVs		gag, pol	SCNT	RNA interference	[78]
		pol2	SCNT	RNA interference	[79]
		pol	SCNT	CRISPR/Cas9	[80]
**Chickens**					
MDV		gB glycoprotein B gene, ICP4	MI	RNA interference	[81]
		ICP4	PGCs	CRISPR/Cas9	[23]
AIV		Virus polymerase	MI	RNA interference	[57]
ALV-J	W38		PGCs	CRISPR/Cas9	[82]
**Cattle**					
FMDV		Viral protein 4	SCNT	RNA interference	[58]
**Sheep**					
Visna virus	Visna virus envelope gene		CMI	DNA construct	[83]

PNI: Pronuclear microinjection; CMI: Cytoplasmic microinjection; SCNT: Somatic cell nuclear transfer; PGC: Primordial germline cells;; FMDV: Foot-and-mouth disease virus; PRRSV: Porcine reproductive and respiratory syndrome virus; CSFV: Classical swine fever virus; TGEV: Transmissible gastroenteritis virus; PDCoV: Porcine deltacoronavirus; PEDV: Porcine epidemic diarrhoea virus; ASFV: African swine fever virus; PERV: Porcine endogenous retroviruses; MDV: Marek’s disease virus; AIV: Avian influenza virus; ALV-J: Avian Leukosis Virus subgroup J.

## Data Availability

No new data were created or analyzed in this study. Data sharing is not applicable to this article.

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
