# Peer review of "Genome Editing Strategies to Protect Livestock from Viral Infections"

_viruses, 2021, doi:10.3390/v13101996_

Round 1
Reviewer 1 Report
In Sollner, Mettenleiter, and Petersen, the authors review potential strategies and current research into the use of CRISPR/Cas to control viral infection in livestock. Such strategies include directly targeting the virus or generating genetically modified animals that are resistant to infection.
The discovery of diverse CRISPR/Cas systems and the ever present risk of infectious diseases that can either spread between animals or spillover into humans provides strong justification for the pursuit of such research and the value of reviewing the current state of this work. However, I urge the authors to think deeply about the organization of this review and what research topics and articles are highlighted. In its current form, this review is challenging to follow as topics are introduced without sufficient background for the journal’s audience and the text which reviews current publications on these CRISPR/Cas applications is often disjointed. Lastly, I believe for this to be a good fit for Viruses, there should be a greater focus on the viruses for which this technology has been applied (Section 5) as this only amounts to about one-third of the text. I have listed below in more detail major and minor suggestions for how to address these concerns.
Major suggestions:
- Organization:
- The introduction and an early section in the review should be focused on what viruses are a disease and economic burden to livestock and what are the current strategies to address these and what are those approaches’ limitations. Topics such as COVID-19 and allocation theory distract from these points. Vaccination is one such approach that is discussed, but others such as quarantining, culling, testing, RNAi (not CRISPR-based), etc. must exist and are not sufficiently discussed. The introduction should also end with a statement about what this review plans to cover.
- Following this refocused introduction, the authors should move to current Section 4. By starting with CRISPR/Cas background it will provide the reader with sufficient background to understand how CRISPR can be used for livestock infection control and introduce useful definitions that are provided in Figure 1 without any context (gene drive, CRISPR screen) and throughout the CRISPR/Cas application sections. In this section, I urge the authors to consider what CRISPR background is necessary. For example, much text is used to define and describe PAMs but this is not a focus in the description of published applications.
- Section 5.1: The paragraphs in this section are quite long and often cover multiple unrelated applications in the same paragraph. The authors should consider breaking these by how Cas is applied e.g. target virus, deliver trans gene, knock out gene, etc. or could organize by virus.
- Missing discussion points/topics areas:
- Discuss the current use of vaccination is commonly used in cattle and in general the benefits of vaccination (currently the text focuses on the challenges)
- As described above, discuss more thoroughly current approaches to manage livestock viral infection and those beyond vaccination – when are they successful, where are they failing (this will strengthen the rationale for why CRISPR/Cas is needed)
- Provide more detail on CRISPR/Cas vaccination strategy and do not separate this into two sections (currently discussed in line 186-190 and line 553-561).
- Discuss diagnostic testing as another approach. Surveillance testing in pigs/livestock is another approach to mitigating infectious disease in livestock (doi: 10.3201/eid2110.140633 among other references) and CRISPR/Cas diagnostics have been developed for viruses that infect livestock such as African Swine Fever virus (doi: doi.org/10.1038/s41421-020-0151-5)
- Expand discussion of the ethics and use of in vivo editing in livestock: delivery approaches, safety of delivery, what are the risks and ethical obstacles.
Minor suggestions:
- Line 73-74: many terms are introduced including maternal antibody block and DIVA which should be more thoroughly explained for the general reader.
- Line 102-103: move CRISPR screens to the discussion as this is not a major focus of the review.
- Line 237 and 247: Cpf1 and C2c2 do not need to be included in the section title. The new names have been readily adopted.
- Line 250: Newer orthologs of Cas13 have been proven to be more effective for RNA targeting work PspCas13b (doi: 10.1126/science.aaq0180) and CasRx (doi: 10.1016/j.cell.2018.02.033)
- Lines 375-395 and section 5.4: These applications seem misplaced since the focus of this section is CRISPR and they instead could be included when introducing of non-CRISPR approaches
- Lines 450-461: a new research area is introduced with limited background
- Section 6.4: missing title
Author Response
Dear Reviewer,
We gratefully acknowledge the reviewing comments on our manuscript ‘CRISPR/Cas strategies to protect livestock from viral diseases’ by Soellner et.al. Please find enclosed our revised version of the manuscript.
We thank you for your valuable and helpful comments and remarks regarding our manuscript. We have in most cases accepted and acted on your suggestions and provide point-by-point explanations below.
Here is our point-by-point explanation to your comments and suggestions.
Major suggestions:
- Comment 1: The introduction and an early section in the review should be focused on what viruses are a disease and economic burden to livestock and what are the current strategies to address these and what are those approaches’ limitations. Topics such as COVID-19 and allocation theory distract from these points. Vaccination is one such approach that is discussed, but others such as quarantining, culling, testing, RNAi (not CRISPR-based), etc. must exist and are not sufficiently discussed. The introduction should also end with a statement about what this review plans to cover.
Thank you for pointing out to focus the introduction on disease, economic burden, and biosecurity. We agree and highlighted current biosecurity measures and their success to control viral diseases. We believe that mentioning sustainable livestock production and the current pandemic put our topic into a broader context relevant for the public. At the end of the introduction, we state the topics covered in the manuscript and relevant search terms.
- Comment 2: Following this refocused introduction, the authors should move to current Section 4. By starting with CRISPR/Cas background it will provide the reader with sufficient background to understand how CRISPR can be used for livestock infection control and introduce useful definitions that are provided in Figure 1 without any context (gene drive, CRISPR screen) and throughout the CRISPR/Cas application sections. In this section, I urge the authors to consider what CRISPR background is necessary. For example, much text is used to define and describe PAMs but this is not a focus in the description of published applications.
We agree to move the section ‘The diversity of CRISPR/Cas’ after the introduction to give the reader a better understanding of the following sections. Also, we reduced the CRISPR/Cas background to the most relevant information to comprehend the topics discussed in the review. Figure 1 was adapted to include CRISPR/Cas diagnostics and vaccine development and moved to the end of the discussion to visualize/summarize CRISPR/Cas strategies to protect livestock from viral disease.
- Comment 3: Section 5.1: The paragraphs in this section are quite long and often cover multiple unrelated applications in the same paragraph. The authors should consider breaking these by how Cas is applied e.g. target virus, deliver trans gene, knock out gene, etc. or could organize by virus.
We have restructured the section according to the applications: Random integration of transgenes, RNA interference, in vivo pathogen genome targeting, elimination of susceptibility.
- Comment 4: Discuss the current use of vaccination is commonly used in cattle and in general the benefits of vaccination (currently the text focuses on the challenges)
We have now included the benefits and development of vaccines throughout the review, for example in the introduction and in ‘CRISPR in disease control and prevention’. We thank you for suggesting discussing vaccine schemes of cattle in our review, however, we think a detailed elaboration would slightly be out of scope for our manuscript. Vaccination against viral diseases is crucial in livestock production but the review focuses on how CRISPR/Cas may be able to enhance current scientific process in understanding viral infections and potentially supplement vaccine schemes. We discuss the effectiveness of PRRSV vaccination as an example and how genome editing, when incorporated in a breeding scheme, can facilitate disease elimination.
- Comment 5: As described above, discuss more thoroughly current approaches to manage livestock viral infection and those beyond vaccination – when are they successful, where are they failing (this will strengthen the rationale for why CRISPR/Cas is needed)
We have adapted the section ‘Strategies to protect livestock from viral infections’ and mention current biosecurity measure. We are glad you have raised the issue to discuss biosecurity and its implementation success in more detail. However, multiple factors can cause biosecurity to fail and in our opinion are too complex to discuss in a short paragraph (see Saegerman et. al. (2012) doi:10.1071/AN15265 or Renault et. al., (2018) doi:10.1111/TBED.12719). We reason that CRISPR/Cas can complement biosecurity and vaccines. This should give the reader a reasonable understanding why CRISPR/Cas can be a rational to manage and understand viral infections.
- Comment 6 + 7: Provide more detail on CRISPR/Cas vaccination strategy and do not separate this into two sections (currently discussed in line 186-190 and line 553-561).
Discuss diagnostic testing as another approach. Surveillance testing in pigs/livestock is another approach to mitigating infectious disease in livestock
Thank you for mentioning possible approaches to develop diagnostic tools and vaccines with CRISPR/Cas. We have now included a section ‘CRISPR in disease control and prevention’ where we discuss both vaccines and diagnostics.
- Comment 8: Expand discussion of the ethics and use of in vivo editing in livestock: delivery approaches, safety of delivery, what are the risks and ethical obstacles.
We agree that genome editing livestock comes with ethical challenges. We included a new section ‘Ethical justification of genome engineering animals for disease resistance’. However, the ethical debate is complex and therefore, we focus in the section on the ethical perspective of generating disease-resistant animals based on a systematic review published in 2019 (de Graeff et al., (2019) doi:10.1098/rstb.2018.0106). Pros and Cons of genome editing for disease resistance are brought forward but also that there is still room for ethical discussion on the topic.
Minor suggestions:
- Comment 1: Line 73-74: many terms are introduced including maternal antibody block and DIVA which should be more thoroughly explained for the general reader
Thank you for pointing out that the terms need further explanation. Maternal antibody block and DIVA vaccines are now explained in more detail.
- Comment 2: Line 102-103: move CRISPR screens to the discussion as this is not a major focus of the review
CRISPR screens are now included in the discussion part of ‘Eliminating susceptibility’ as method to identify target genes.
- Comment 3: Line 237 and 247: Cpf1 and C2c2 do not need to be included in the section title. The new names have been readily adopted
We have excluded the old terminology in the section title.
- Comment 4: Line 250: Newer orthologs of Cas13 have been proven to be more effective for RNA targeting work PspCas13b (doi: 10.1126/science.aaq0180) and CasRx (doi: 10.1016/j.cell.2018.02.033)
We agree that more effective Cas13 proteins for RNA editing have been brought forward and the new references are now included.
- Comment 5: Lines 375-395 and section 5.4: These applications seem misplaced since the focus of this section is CRISPR and they instead could be included when introducing of non-CRISPR approaches
The lines are now placed in the new section of ‘RNA interference’
- Comment 6: Lines 450-461: a new research area is introduced with limited background
An introductory sentence for GWAS is now included in the section.
- Comment 7: Section 6.4: missing title
A title is now added to the section
We are looking forward to hearing from you.
Sincerely yours,
Björn Petersen
Reviewer 2 Report
The review proposed by Sollner and colleagues aimed to report the studies that use CRISPR/Cas techniques to control viral infection in domestic animals.
The review is very exhaustive and the bibliographic revision is well conducted.
The text contains several typos and grammatical errors, please revised it deeply.
Although it is a review, the M&M or the bibliography research criteria must be reported in the text.
Author Response
Dear Reviewer,
We gratefully acknowledge your reviewing comments on our manuscript ‘CRISPR/Cas strategies to protect livestock from viral diseases’ by Soellner et.al. Please find enclosed our revised version of the manuscript.
We thank you for your valuable and helpful comments on our manuscript. We have acted on your minor suggestions and provide point-by-point explanations below. We have also incorporated major changes in the structure and discuss more content on vaccine and virus diagnostic based on CRISPR/Cas based on the suggestions by reviewer 1.
Minor suggestions:
- Comment 1: The text contains several typos and grammatical errors, please revised it deeply
We have revised the text and hope that all spelling and grammatical errors are eradicated.
- Comment 2: Although it is a review, the M&M or the bibliography research criteria must be reported in the text
We included search terms we have used for literature research in the introduction.
We are confident we have addressed the issues raised by you and thereby increased the quality of the revised manuscript to make it acceptable for publication in Viruses.
Sincerely yours,
Björn Petersen
Round 2
Reviewer 1 Report
I applaud the authors for sharing this brilliantly revised review article. I thank the authors for thoroughly addressing my concerns which meant quite a bit of restructuring of the text as well as adding new concepts. I believe this review article is now ready for publication, but I do want to make one very minor suggestion: I believe the addition of Cas3 mentioned in lines 155-157 is interesting given its recent use, so it is worth mentioning briefly in these lines. However, I believe the entire subsection 2.4 about Cas3 is not necessary (lines 506-511) since this Cas protein has yet to be applied within the livestock field and is thus not mentioned or discussed in later sections.